# Nor-24-homoscalaranes, Neutrophilic Inflammatory Mediators from the Marine Sponge *Lendenfeldia* sp.

**DOI:** 10.3390/ph16091258

**Published:** 2023-09-06

**Authors:** Bo-Rong Peng, Li-Guo Zheng, Lo-Yun Chen, Mohamed El-Shazly, Tsong-Long Hwang, Jui-Hsin Su, Mei-Hsien Lee, Kuei-Hung Lai, Ping-Jyun Sung

**Affiliations:** 1Graduate Institute of Pharmacognosy, College of Pharmacy, Taipei Medical University, Taipei 110301, Taiwan; peng_br@tmu.edu.tw (B.-R.P.); m303110004@tmu.edu.tw (L.-Y.C.); lmh@tmu.edu.tw (M.-H.L.); 2Doctoral Degree Program in Marine Biotechnology, National Sun Yat-sen University, Kaohsiung 804201, Taiwan; t0919928409@gmail.com; 3National Museum of Marine Biology and Aquarium, Pingtung 944401, Taiwan; x2219@nmmba.gov.tw; 4PhD Program in Clinical Drug Development of Herbal Medicine, College of Pharmacy, Taipei Medical University, Taipei 110301, Taiwan; 5Department of Pharmacognosy, Faculty of Pharmacy, Ain-Shams University, Organization of African Unity Street, Abassia, Cairo 11566, Egypt; mohamed.elshazly@pharma.asu.edu.eg; 6Research Center for Chinese Herbal Medicine, College of Human Ecology, Chang Gung University of Science and Technology, Taoyuan 333324, Taiwan; htl@mail.cgu.edu.tw; 7Graduate Institute of Health Industry Technology, College of Human Ecology, Chang Gung University of Science and Technology, Taoyuan 333324, Taiwan; 8Graduate Institute of Natural Products, College of Medicine, Chang Gung University, Taoyuan 333323, Taiwan; 9Department of Chemical Engineering, Ming Chi University of Technology, New Taipei City 243303, Taiwan; 10Department of Anesthesiology, Chang Gung Memorial Hospital, Taoyuan 333423, Taiwan; 11Department of Marine Biotechnology and Resources, National Sun Yat-sen University, Kaohsiung 804201, Taiwan; 12Center for Reproductive Medicine and Sciences, Taipei Medical University Hospital, Taipei 110301, Taiwan; 13Traditional Herbal Medicine Research Center, Taipei Medical University Hospital, Taipei 110301, Taiwan; 14Graduate Institute of Natural Products, Kaohsiung Medical University, Kaohsiung 807378, Taiwan; 15Chinese Medicine Research and Development Center, China Medical University Hospital, Taichung 404394, Taiwan; 16PhD Program in Pharmaceutical Biotechnology, Fu Jen Catholic University, New Taipei City 242062, Taiwan

**Keywords:** marine sponge, *Lendenfeldia* sp., scalarane, anti-inflammatory, neutrophil

## Abstract

The marine sponge *Lendenfeldia* sp., collected from the Southern waters of Taiwan, was subjected to chemical composition screening, resulting in the isolation of four new 24-homoscalarane compounds, namely lendenfeldaranes R–U (**1**–**4**). The structures and relative stereochemistry of the new metabolites **1**–**4** were assigned based on NMR studies. The absolute configurations of compounds **1**–**4** were determined by comparing the calculated and experimental values of specific optical rotation. The antioxidant and anti-inflammatory activities of the isolated compounds were assayed using superoxide anion generation and elastase release assays. These assays are used to determine neutrophilic inflammatory responses of respiratory burst and degranulation. Compounds **2** and **4** inhibited superoxide anion generation by human neutrophils in response to formyl-L-methionyl-L-leucyl-L-phenylalanine/cytochalasin B (fMLP/CB) with IC_50_: 3.98–4.46 μM. Compounds **2** and **4** inhibited fMLP/CB-induced elastase release, with IC_50_ values ranging from 4.73 to 5.24 μM. These findings suggested that these new 24-homoscalarane compounds possess unique structures and potential anti-inflammatory activity.

## 1. Introduction

Marine sponges belonging to the genus *Lendenfeldia* (phylum Porifera, class Demospongia, subclass Keratosa, order Dictyoceratida, family Thorectidae) are widely distributed in the flat reefs of the Asia–Pacific region. These sponges can also be found in aquarium tanks and are considered pests as they can proliferate in aquacultures even under standard conditions [1]. Previous research on *Lendenfeldia* sponges revealed a rich diversity of secondary metabolites, with sesterterpenoids being particularly prominent [2,3,4,5,6,7,8,9]. These sesterterpenoids exhibit a broad spectrum of biological activities, including cytotoxicity [10,11,12,13,14,15,16,17,18,19,20,21,22,23,24,25,26,27,28,29,30,31], anti-inflammatory properties [5,7,18,32,33,34,35,36], anti-HIV effects [3,37], anti-microbial [15,16,22,38,39,40,41] and anti-neurofibroma activity [42]. Anti-neutrophil inflammatory agents refer to substances that aim to reduce or inhibit the inflammatory response mediated by neutrophils. Neutrophils are a type of white blood cell that are essential to the immune response. They play a crucial role in the initial response to infection and tissue injury [43]. While neutrophils are important for combating pathogens and promoting tissue repair, their excessive or prolonged activation can lead to harmful inflammation and tissue damage. Therefore, targeting neutrophil-mediated inflammation is an area of interest in various inflammatory conditions, including autoimmune diseases, acute respiratory distress syndrome (ARDS), and inflammatory bowel disease (IBD) [44]. In our two previous reports, we isolated a series of scalarane-type sesterterpenoids from the marine sponge *Lendenfeldia* sp. and tested their anti-inflammatory activity against superoxide anion generation and elastase release, which represented the neutrophilic inflammatory responses of respiratory burst and degranulation, respectively. The results indicated that the potent activity of this class of compounds could lead to their further development as anti-neutrophilic agents [5,7]. In our ongoing investigation, aiming to discover new metabolites from *Lendenfeldia* sp., we identified four novel 24-homoscalarane-type sesterterpenoids named lendenfeldaranes R–U (**1**–**4**) (Figure 1). The determination of the structures of compounds **1**–**4** involved a thorough analysis of their Infrared (IR), Specific Optical Rotation (SOR), Mass (MS), and Nuclear Magnetic Resonance (NMR) spectra. Moreover, their NMR data were compared with those of known compounds with structural similarities. To establish the absolute configuration of compounds **1**–**4**, we compared the experimental optical rotation with the calculated optical rotation spectra. Additionally, we evaluated the anti-inflammatory activity of these compounds by assessing their ability to inhibit superoxide anion generation and elastase release in N-formyl-methionyl-leucyl phenylalanine/cytochalasin B (fMLF/CB)-induced human neutrophils.

## 2. Results and Discussion

Compound **1** was obtained as a white powder, and its molecular formula was determined as C_27_H_42_O_5_ based on the presence of a sodium adduct at *m*/*z* 469.29256 in HRESIMS (calculated for C_27_H_42_O_5_ + Na, 469.29245) and ^13^C NMR data (Table 1). The IR spectrum of **1** exhibited characteristic peaks indicating the presence of ketone carbonyl (ν_max_ 1703 cm^−1^), ester carbonyl (ν_max_ 1736 cm^−1^), and hydroxy (ν_max_ 3429 cm^−1^) functional groups. The ^1^H NMR spectroscopic data of **1** (Table 1) revealed six methyl groups at δ_H_ 0.85, 0.88, 1.13, 1.26, 2.06, and 2.19 (each 3H, singlet), and one oxymethine proton at δ_H_ 3.80 (1H, triple doublet, *J* = 9.6, 9.6, 3.2 Hz). The presence of an oxymethylene group was indicated by the anisochronous signals of the geminal protons observed at δ_H_ 4.67 (1H, doublet, *J* = 12.4 Hz) and 4.21 (1H, doublet of doublets, *J* = 12.4, 1.2 Hz). An analysis of the HSQC and ^13^C spectroscopic data revealed that compound **1** consists of 27 carbon atoms, including six methyl groups, nine sp^3^ methylene groups (including one oxymethine), five sp^3^ methine groups (including one oxymethine), four sp^3^ quaternary carbons, and three sp^2^ quaternary carbons (including three carbonyls). Based on the ^1^H and ^13^C NMR spectroscopic data, compound **1** was found to possess an acetoxy group (δ_H_ 2.06, 3H, singlet; δ_C_ 170.8, carbonyl; 21.1, methyl) and two ketone carbonyls (δ_C_ 212.4, carbonyl; 214.4, carbonyl), accounting for three degrees of unsaturation. The remaining four degrees of unsaturation indicated that compound **1** possesses a tetracyclic structure.

To gain a more comprehensive understanding of the structure of **1**, 2D NMR spectra, including ^1^H−^1^H correlation spectroscopy (COSY) and heteronuclear multiple-bond coherence (HMBC) spectra were employed. These spectra were used to analyze the correlations between protons (^1^H) and neighboring atoms, as well as the connectivity of multiple bonds between different nuclei (Figure 2). The ^1^H−^1^H COSY cross-peaks revealed four distinct proton spin systems: H-5/H_2_-6/H_2_-7, H_2_-1/H_2_-2/H_2_-3, H-9/H_2_-10, and H-14/H_2_-15/H-16/H-17/H_2_-18. The HMBC cross-signals between H_3_-21 and C-7, C-8, C-9, C-14; H_3_-20 and C-3, C-4, C-5, C-19; H_2_-22 and C-1, C-9, the ester carbonyl (δ_C_ 170.8); H_3_-23 and C-12, C-13, C-14, C-18; H_3_-25 and C-17, C-24; H_2_-11 and C-12 established connections between the partial structures. A comparison with the previous literature aided in successfully constructing the overall structure of **1**. The obtained data showed similarities to a known 24-homescalarane compound, felixin F (**5**) [45]. However, notable differences were observed between compounds **1** and **5**. In compound **5**, the hydroxy group at C-22 (δ_H_ 3.93, 1H, doublet, *J* = 11.6 Hz; 4.07, 1H, doublet, *J* = 11.6 Hz; δ_C_ 62.7) was replaced by an acetoxy group in compound **1** (δ_H_ 4.67, 1H, doublet, *J* = 12.4 Hz; 4.21, 1H, doublet of doublets, J = 12.4, 1.2 Hz; δ_C_ 64.5). Additionally, **1** lacked the aldehyde group present in compound **5**, which was substituted by a methylene carbon.

Through a literature search, it was found that all naturally occurring scalaranes exhibit β-oriented methyl groups (Me-23 and Me-22) at C-13 and C-10, respectively, regardless of their oxidation state (-CH_2_OH, -CH_2_OAc, -COOH, and -CHO) [7]. The orientations of these methyl groups remained consistent within the carbon skeleton, with C-13 and C-10 serving as anchor points for analysis. The relative configuration of **1** was established through NOESY experiments (Figure 3). The NOESY interactions between H_3_-23 and H-17, as well as H_3_-21, and between H_2_-22 and H_3_-20, and H_3_-21, revealed the β-orientation of H_3_-23, H_2_-22, H_3_-21, H_3_-20, and H-17. An interaction between H-5 and H-9 indicated their spatial proximity. A further correlation analysis showed a correlation between H-9 and H-14, as well as a correlation between H-14 and H-16, suggesting an α-orientation for H-5, H-9, H-14, and H-16. Based on these findings, the structure of **1** was determined, and the assigned stereogenic centers were identified as (5*S**,8*R**,9*S**,10*R**,13*R**,14*S**, 16*S**,17*S**).

To determine the absolute stereochemistry of **1**, two possible configurations were considered: **1**-5*S*,8*R*,9*S*,10*R*,13*R*,14*S*,16*S*,17*S*, and **1**-5*R*,8*S*,9*R*,10*S*,13*S*,14*R*,16*R*,17*R*. These configurations were fed into Gaussian 16 software to calculate the conformation, optimize the structure, and determine the specific optical rotation (SOR) values (Table 2). The calculated SOR value of **1**-5*S*,8*R*,9*S*,10*R*,13*R*,14*S*,16*S*,17*S* (+31) was consistent with the experimental result of compound **1** (positive). Based on these results, the configurations of the stereogenic centers in **1** were determined to be (5*S*,8*R*,9*S*,10*R*,13*R*,14*S*,16*S*,17*S*). Consequently, the structure of **1** was identified as a new sesterterpenoid and named lendenfeldarane R.

Compound **2** was discovered to have the molecular formula C_25_H_38_O_4_, which was determined from a (+)-HRESIMS signal at *m*/*z* 425.26645 (calculated for C_25_H_38_O_4_+Na, 425.26623) and ^13^C data indicating the presence of seven unsaturated degrees. The IR spectrum of **2** showed absorption peaks for carbonyl groups (ν_max_ 1707 cm^−1^ and 1676 cm^−1^) and a hydroxy group (ν_max_ 3417 cm^−1^). Analyzing the 1D NMR data (Table 2), it was found that compound **2** was similar to felixin B (**6**) [46], with the main difference being the presence of a functional group at C-12. In compound **2**, the chemical shift of H-12 (δ_H_ 3.98, 1H, br s) was shifted downfield compared to its counterpart in felixin B (**6**) (δ_H_ 4.97, 1H, dd, J = 2.8, 2.8 Hz). The absence of acetoxy signals suggested that felixin B (**6**) was the 12-acetyl derivative of compound **2**. Further confirmation of the planar structure of compound **2** was obtained through the interpretation of 2D NMR spectroscopic data (Figure 2). The correlations observed among the chiral centers in the core rings A-D of **2** were identical to those observed in **1**. In the NOESY experiment of **2** (Figure 3), the α-orientation of the hydroxy group at C-12 was determined based on the NOESY correlation between H-12 and H_3_-23. As a result, the configurations of the stereogenic carbons in **2** were established as (5*S**,8*R**,9*S**,10*R**,12*S**, 13*R**,14*S**). The SOR was employed to determine the absolute configuration of **2**. The calculated SOR values for **2**-5*S*,8*R*,9*S*,10*R*,12*S*,13*R*,14*S* and **2**-5*R*,8*S*,9*R*,10*S*,12*R*,13*S*,14*R* were positive (78) and negative (−78), respectively (Table 2). The experimental SOR data of **2** (positive) matched with the configuration **2**-5*S*,8*R*,9*S*,10*R*,12*S*,13*R*,14*S*. Based on the aforementioned analyses, the structure of **2** was successfully determined, leading to its identification and designation as lendenfeldarane S.

Compound **3** was acquired in the form of a white powder, and its molecular formula was determined as C_27_H_42_O_6_ based on the HRESIMS signal at *m*/*z* 485.28737 [M + Na]^+^ (calculated for C_27_H_42_O_6_ + Na, 485.28736) and ^13^C data suggesting the presence of seven degrees of unsaturation. The IR spectrum of **3** exhibited prominent peaks at ν_max_ 3459, 1727, and 1666, indicating the presence of hydroxy, ester, and α,β-unsaturated ketone groups, respectively. Analysis of the ^1^H and ^13^C NMR spectroscopic data (Table 3) revealed the presence of an acetoxy group (δ_H_ 2.03, 3H, s; δ_C_ 170.8, C; 21.3, CH_3_) and a ketonic carbonyl group (δ_C_ 199.3) within compound **3**. Moreover, the ^13^C resonances at δ_C_ 133.6 (C) and 155.7 (CH) suggested the existence of a trisubstituted olefin, accounting for three degrees of unsaturation. Compound **3** was identified as an analog of tetracyclic sesterterpenoid. The NMR data resembled those of felixin C (**7**) [46]. The chemical shift of H-16 in felixin C (**7**) (δ_H_ 4.55, d, *J* = 4.8 Hz) was shifted downfield in **3** (δ_H_ 4.91, dd, *J* = 4.0, 1.6 Hz), with an additional signal for a hydroxyperoxide group (δ_H_ 9.27, 1H, s). The presence of a hydroperoxide group substitution at position 16 was deduced from the HMBC cross-peak (Figure 2) between H-16 and C-14, C-17, C-18, and from H-14 to C-16, as well as from the COSY correlation between H-15, H-14, and H-16. Additionally, comparing the ^13^C NMR data with those of **1** and **4**, the C-16 carbon resonating at *δ*_C_ 77.2 in **3** was more downfield than δ_C_ 70.9 in **1** and δ_C_ 68.1 in **4**, revealing that the hydroxyperoxide group was located at C-16 position.

Further insights into the structure of **3** were obtained through the NOESY experiment (Figure 3). The observed correlations provided valuable information regarding the configurations of the chiral centers in the core rings A-C of **3**, which were found to be identical to those observed in **1**. Notably, H_3_-23 showed NOE correlations with H_3_-21 and H-12, indicating the β-orientation of H-12. Additionally, H-14 displayed NOE correlations with H-9 and H-16, indicating the β-orientation of the hydroperoxide group at C-16. To determine the absolute configuration of **3**, a comparison was made between its experimental optical rotation and the corresponding SOR value. The calculated SOR values for **3**-5*S*,8*R*,9*S*,10*R*,12*S*,13*R*,14*S*,16*S* and **3**-5*R*,8*S*,9*R*,10*S*,12*R*,13*S*,14*R*,16*R* were positive (15) and negative (−15), respectively (Table 2). The experimental SOR data of **3** (positive) matched with the configuration **3**-5*S*,8*R*,9*S*,10*R*,12*S*,13*R*,14*S*,16*S*. Based on the aforementioned results, the structure of **3** was determined and assigned the name lendenfeldarane T.

Compound **4** was obtained in the form of an unstructured fine powder. Its molecular formula was determined to be C_27_H_42_O_5_ based on the (+)-HRESIMS pseudo-molecular ion peak at *m*/*z* 469.29221 (calculated for C_27_H_42_O_5_+Na, 469.29245) and ^13^C data, which indicated the presence of seven degrees of unsaturation. The IR spectrum of **4** exhibited absorption bands corresponding to hydroxy groups (maximum absorption at 3486 cm^−1^), ester carbonyl groups (maximum absorption at 1726 cm^−1^), and α,β-unsaturated ketone groups (maximum absorption at 1655 cm^−1^). The structure of **4** was determined through the analysis of 1D and 2D NMR spectroscopic data (refer to Table 3). Based on these findings, it was observed that the overall structure of **4** closely resembled that of felixin B (**7**) [46]. The ^13^C and ^1^H NMR data of **4** were similar to those of compound **7**, except for carbon resonances at C-16 and C-17, which appeared at δ_C_ 63.3 (CH) and 138.2 (C) in compound **7** and at δ_C_ 68.1 (CH) and 134.2 (C) in **4**, indicating that **4** was the 16S isomer of compound **7**. Analysis of the NOESY spectra (refer to Figure 3) revealed a cross-peak from H-14 to H-16, indicating an α-orientation of H-16 in **4**. According to the above data, the structure of **4** was determined, and the compound was named lendenfeldarane U.

*N*-formyl-methionyl-leucyl-phenylalanine (fMLF) and pathogen-associated molecular patterns (PAMPs) can stimulate neutrophils, leading to the initiation of inflammatory responses such as the generation of O_2_^•−^ (respiratory burst) and the release of elastase (degranulation) [7]. To evaluate the anti-inflammatory properties of nor-24-homoscalaranes, all the isolated compounds were tested on fMLF-induced human neutrophils, and the findings are summarized in Table 4. As a positive control, LY294002, a phosphoinositide 3-kinase (PI3K) inhibitor, was employed, given the established role of PI3K in regulating neutrophil respiratory burst and/or degranulation [47,48]. Compounds **2** and **4** demonstrated significant activity against both O_2_^•−^ accumulation (IC_50_ = 3.98–4.46 μM) and elastase release (IC_50_ = 4.73–5.24 μM). On the other hand, compounds **1** and **3** were inactive at a concentration of 10 μM. These results emphasized the importance of the conjugated functionality at C-17-18-24 for anti-inflammatory activity, while the substitution of the peroxyl group at C-16 may diminish this effect.

## 3. Materials and Methods

### 3.1. General Experimental Procedures

IR spectra were recorded using a Thermo Scientific Nicolet iS5 FT-IR spectrophotometer (Thermo Scientific, Waltham, MA, USA). Optical activities were measured using a JASCO P-1010 polarimeter (JASCO, Tokyo, Japan). NMR spectra were obtained using JEOL ECZ 400S or 600R NMR spectrometers (JEOL, Tokyo, Japan) with CDCl_3_ (Sigma-Aldrich, St. Louis, MO, USA) as the deuterated solvent. The detected signals in ^1^H and ^13^C NMR were corrected at 7.26 ppm (singlet) and 77.0 ppm (triplet), respectively. The coupling constants (*J*) were converted to Hz. MS data, including ESIMS and HRESIMS, were obtained using a Bruker 7 Tesla solera FTMS system (Bruker, Bremen, Germany). Two types of TLC analyses were performed using aluminum plates coated with Kieselgel 60 F_254_ (0.25 mm) and RP-18 F_254S_ (0.25 mm) (Merck, Darmstadt, Germany). For chromatographic separation, a glass column was used, which was packed with a stationary phase of silica gel 60 (40–63 μm and 63–200 μm, Merck, Darmstadt, Germany). NP-HPLC was conducted using a system consisting of a solvent delivery system (L-7110, Hitachi, Tokyo, Japan) and a preparative packed silica gel column (YMC-Pack SIL, SIL-06, 250 × 20 mm, D. S-5 μm) (Sigma-Aldrich, St. Louis, MO, USA). RP-HPLC was performed using a system consisting of a solvent delivery system (L-2130, Hitachi, Tokyo, Japan), a PDA detector (L-2455, Hitachi, Tokyo, Japan), and a C_18_ column (Luna 5 μm, C_18_(2) 100Å AXIA Packed, 250 × 21.2 mm) (Phenomenex, Torrance, CA, USA).

### 3.2. Animal Material and Isolation of Compounds

The *Lendenfeldia* sp. specimen was obtained by scuba diving off the coast of Southern Taiwan in April 2019. At the National Museum of Marine Biology & Aquarium, Taiwan, a specimen with a voucher number (specimen No. 2019-04-SP) was deposited. *Lendenfeldia* sp. was taxonomically identified by Prof. Yusheng M. Huang from the National Penghu University of Science and Technology, Taiwan. *Lendenfeldia* sp. was collected (2.9 kg) and freeze-dried. The freeze-dried material (213 g, dry weight) was minced and an extract was prepared using a 1:1 mixture of CH_2_Cl_2_:MeOH (1 L × 6). The resulting extract underwent liquid–liquid partitioning between EtOAc and H_2_O. The obtained EtOAc layer (7.9 g) was further subjected to normal-phase column chromatography. Elution was carried out using a gradient solvent system comprising *n*-hexane, followed by increasing polarity mixtures of *n*-hexane and EtOAc, pure acetone, and finally pure methanol as eluting solvents. This process resulted in the production of 14 sub-fractions labeled A–N. Sub-fraction H underwent reversed-phase chromatography on C_18_ silica gel, eluted with a MeOH:H_2_O mixture (50% MeOH to pure MeOH), and yielded six additional sub-fractions, H1–H6. Sub-fraction H5 then underwent RP-HPLC using an isocratic solvent system of MeOH:H_2_O (8:2), resulting in the isolation of compound **2** (1.3 mg). Fraction I was purified using NP-HPLC using a CH_2_Cl_2_:acetone mixture (4:1) as an isocratic solvent system with a flow rate of 3.0 mL/min, yielding eight sub-fractions, which were labeled I1–I8. Sub-fraction I5 was further purified using RP-HPLC with an isocratic solvent system of MeOH:H_2_O (4:1) at a flow rate of 5 mL/min, resulting in the isolation of compounds **3** (0.9 mg) and **4** (2.1 mg). Similarly, sub-fraction I6 underwent RP-HPLC using an isocratic solvent system of MeOH:H_2_O (4:1) at a flow rate of 5 mL/min, leading to the isolation of compound **1** (0.5 mg).

Lendenfeldarane R (**1**): Amorphous powder; ${\left[\alpha \right]}_{D}^{25}$ +21 (*c* 0.025, CHCl_3_); IR (ATR) *ν*_max_ 3429, 1736, 1703 cm^−1^; ESIMS *m*/*z* 469 [M + Na]^+^; HRESIMS *m*/*z* 469.29256 [M + Na]^+^ (calcd. for C_27_H_42_O_5_ + Na, 469.29245); ^1^H (400 MHz, CDCl_3_) and ^13^C (100 MHz, CDCl_3_) NMR spectroscopic data.

Lendenfeldarane S (**2**): Amorphous powder; ${\left[\alpha \right]}_{D}^{25}$ +61 (*c* 0.025, CHCl_3_); IR (ATR) *ν*_max_ 3417, 1707, 1676 cm^−1^; ESIMS *m*/*z* 425 [M + Na]^+^; HRESIMS *m*/*z* 425.26645 [M + Na]^+^ (calcd. for C_25_H_38_O_4_ + Na, 425.26623); ^1^H (600 MHz, CDCl_3_) and ^13^C (150 MHz, CDCl_3_) NMR spectroscopic data.

Lendenfeldarane T (**3**): Amorphous powder; ${\left[\alpha \right]}_{D}^{25}$ +40 (*c* 0.045, CHCl_3_); IR (ATR) *ν*_max_ 3459, 1727, 1666 cm^−1^; ESIMS *m*/*z* 485 [M + Na]^+^; HRESIMS *m*/*z* 485.28737 [M + Na]^+^ (calcd. for C_27_H_42_O_6_ + Na, 485.28736); ^1^H (400 MHz, CDCl_3_) and ^13^C (100 MHz, CDCl_3_) NMR spectroscopic data.

Lendenfeldarane U (**4**): Amorphous powder; ${\left[\alpha \right]}_{D}^{25}$ +76 (*c* 0.105, CHCl_3_); IR (ATR) *ν*_max_ 3486, 1726, 1655 cm^−1^; ESIMS *m*/*z* 469 [M + Na]^+^; HRESIMS *m*/*z* 469.29221 [M + Na]^+^ (calcd. for C_27_H_42_O_5_ + Na, 469.28736); ^1^H (600 MHz, CDCl_3_) and ^13^C (150 MHz, CDCl_3_) NMR spectroscopic data.

### 3.3. In Silico Calculations

To minimize molecular energy, the molecular structures were optimized at the MM2 level, resulting in the generation of a mol file. This mol file was analyzed using the MMFF94 force field in GaussView 6.1 (Gaussian Inc., Wallingford, CT, USA) with the assistance of the GMMX package, which allowed us to explore the conformational search results. The obtained data were imported into Gaussian 16 software (Gaussian Inc., Wallingford, CT, USA). In Gaussian 16, the structures underwent further optimization using the time-dependent density functional theory (TDDFT) methodology. The optimization was conducted in the solvent phase using the PCM/mpw1pw91/6-31 g(d,p) levels, enabling GIAO-DFT calculation. In the final step, the computed NMR results were averaged, taking into consideration the proportion of each conformer [49].

### 3.4. Preparation of Human Neutrophils

Blood samples were collected via venipuncture from human donors aged between 20 and 30 years. The Institutional Review Board (IRB) of Chang Gung Memorial Hospital approved and oversaw the protocol, which was identified as IRB No. 202002493A3. Neutrophil purification was performed using a previously established technique [7]. The process involved several steps, including hypotonic lysis, dextran sedimentation, and separation of erythrocytes using a Ficoll Hypaque gradient. Once the human neutrophils were isolated, they were placed in a 50 mL centrifuge tube containing an HBSS buffer solution devoid of calcium (Ca^2+^) and magnesium (Mg^2+^). The pH of the solution was adjusted to 7.4, and the viability of the neutrophils was assessed using the trypan blue exclusion method to ensure that more than 98% of the cells remained viable. Subsequently, the neutrophils were examined in HBSS with 1 mM CaCl_2_ at a temperature of 37 °C.

### 3.5. Measurement of Superoxide Anion (O_2_^•−^) Generation

The evaluation of O_2_^•−^ generation involved the utilization of ferricytochrome *c* and the inhibitory effect of superoxide dismutase (SOD) on its reduction process [15]. After adding ferricytochrome *c* (0.6 mg/mL), neutrophils at a concentration of 6 × 10^5^ cells/mL were equilibrated at 37 °C and incubated for 5 min. Subsequently, the neutrophils were treated with either pure compounds or DMSO (0.1% as a control). To enhance the reaction, cytochalasin B (CB) was introduced at a concentration of 1 μg/mL [7]. The activated mixture was then incubated for 3 min after stimulation with 0.1 μM fMLF. The reduction of ferricytochrome *c* was continuously monitored at 550 nm by measuring absorbance changes using a spectrophotometer (U-3010, Hitachi, Tokyo, Japan).

### 3.6. Measurement of Elastase Release

To evaluate the degranulation of azurophilic granules, an elastase release assay was conducted using MeO-Suc-Ala-Ala-Pro-Val-p-nitroanilide as the substrate for elastase [7]. Neutrophils (6 × 10^5^ cells/mL) were equilibrated at 37 °C after the addition of MeO-Suc-Ala-Ala-Pro-Val-p-nitroanilide (100 μM). The cells were then incubated for 5 min before treatment with pure compounds. To enhance the reaction, CB (0.5 g/mL) was added, followed by the introduction of fMLF (0.1 μM) to induce cellular activation. Elastase release was assessed by continuously monitoring changes in absorbance at 405 nm.

### 3.7. Statistics

Statistical calculations were performed using Student’s *t*-test (GraphPad Software 9.0.2, San Diego, CA, USA). A significance level of *p* < 0.05 was employed to determine statistical significance.

## 4. Conclusions

In this study, we identified and characterized four new 24-homoscalarane compounds, lendenfeldaranes R–U (**1**–**4**), isolated from *Lendenfeldia* sp. through comprehensive chromatographic and spectroscopic analyses. The structures, relative stereochemistry, and absolute configurations of these compounds were determined. The in vitro assays demonstrated that compounds possessing C-17-18-24-conjugated functionality exhibited significant inhibitory effects on key inflammatory responses, including the generation of superoxide anions and the release of elastase in human neutrophils. Our findings showed that *Lendenfeldia* sp. is an excellent source of anti-inflammatory agents with unique structures that can be further developed into potential drug leads.

## Figures and Tables

**Figure 1 pharmaceuticals-16-01258-f001:**
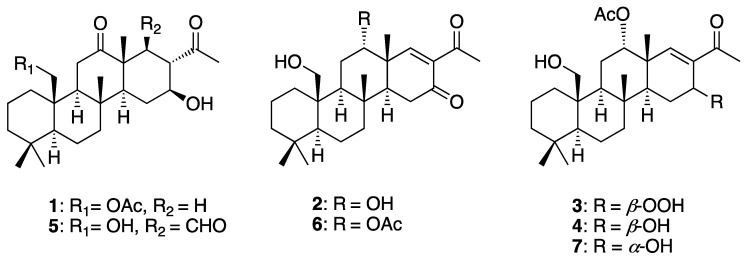
The identified 24-homoscalaranes from the marine sponge *Lendenfeldia* sp.

**Figure 2 pharmaceuticals-16-01258-f002:**
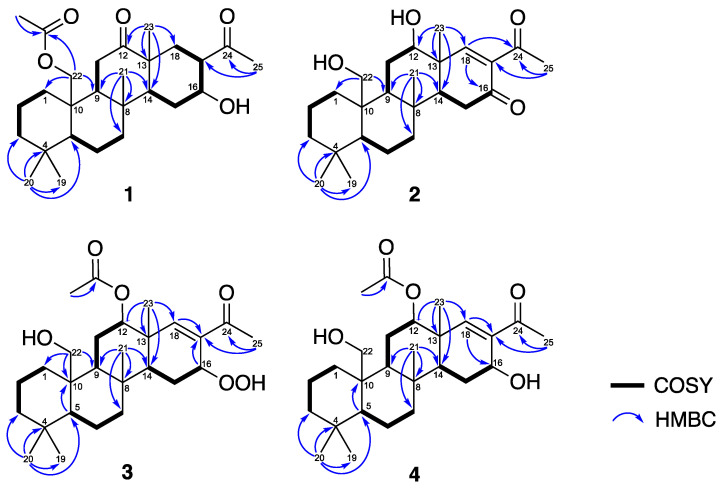
Key 2D correlations of planar structures of **1**–**4**.

**Figure 3 pharmaceuticals-16-01258-f003:**
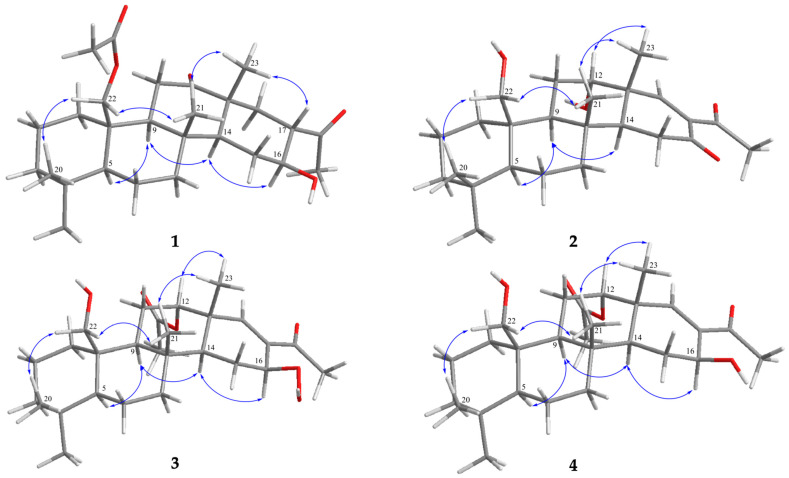
Selective NOESY correlations (
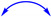
) of relative structures of **1**–**4**.

**Table 1 pharmaceuticals-16-01258-t001:** The NMR data for compounds **1** and **2**.

	1		2	
C/H	*δ*_H_ (*J* in Hz) ^a^	*δ*_C_ (mult.) ^b^	*δ*_H_ (*J* in Hz) ^c^	*δ*_C_ (mult.) ^d^
1	2.00 m; 0.77 ddd (12.4, 12.4, 4.4) ^e^	34.3, CH_2_ ^f^	2.13 m; 0.73 ddd (12.0, 12.0, 3.0) ^e^	34.1, CH_2_ ^f^
2	1.61 m; 1.48 m	17.9, CH_2_	1.56 m; 1.37 m	17.7, CH_2_
3	1.45 m; 1.17 m	41.4, CH_2_	1.43 m; 1.19 m	41.7, CH_2_
4		33.0, C		33.0, C
5	1.00 m	56.8, CH	1.03 dd (12.6, 2.4)	56.9, CH
6	1.61 m	18.2, CH_2_	1.49 m	18.4, CH_2_
7	1.95 m; 1.15 m	41.5, CH_2_	1.77 ddd (12.6, 3.0, 3.0); 1.08 m	41.1, CH_2_
8		37.2, C		37.3, C
9	1.32 m	60.2, CH	1.52 m	51.8, CH
10		41.0, C		41.8, C
11	3.97 dd (14.4, 14.4) 2.50 dd (14.4, 2.8)	37.6, CH_2_	2.32 m; 1.86 ddd (16.2, 2.4, 2.4)	28.3, CH_2_
12		214.4, C	3.98 br s	73.9, CH
13		48.5, C		42.6, C
14	1.19 m	55.9, CH	2.14 m	47.5, CH
15	1.98 m; 1.57 m	27.6, CH_2_	2.53 m; 2.24 m	34.9, CH_2_
16	3.80 ddd (9.6, 9.6, 3.2)	70.9, CH		198.2, C
17	2.49 m	53.7, CH		142.8, C
18	2.05 m	36.8, CH	7.55 s	165.9, CH
19	0.88 s	33.7, CH_3_	0.86 s	33.9, CH_3_
20	0.85 s	21.8, CH_3_	0.77 s	21.8, CH_3_
21	1.13 s	16.0, CH_3_	1.11 s	15.8, CH_3_
22	4.67 d (12.4) 4.21 dd (12.4, 1.2)	64.5, CH_2_	4.06 d (11.4); 3.09 dd (11.4)	62.8, CH_2_
23	1.26 s	19.3, CH_3_	1.11 s	18.6, CH_3_
24		212.4, C		198.3, C
25	2.19 s	28.8, CH_3_	2.45 s	30.7, CH_3_
22-OAc		170.8, C		
	2.06 s	21.1, CH_3_		

^a^ Spectra recorded at 400 MHz in CDCl_3_; ^b^ spectra recorded at 100 MHz in CDCl_3_; ^c^ spectra recorded at 600 MHz in CDCl_3_; ^d^ spectra recorded at 150 MHz in CDCl_3_; ^e^ *J* values (in Hz) in parentheses; ^f^ attached protons were deduced by the HSQC experiment.

**Table 2 pharmaceuticals-16-01258-t002:** The NMR data for compounds **3** and **4**.

	3		4	
C/H	*δ*_H_ (*J* in Hz) ^a^	*δ*_C_ (mult.) ^b^	*δ*_H_ (*J* in Hz) ^c^	*δ*_C_ (mult.) ^d^
1	2.08 m; 0.58 ddd (13.2, 13.2, 2.8) ^e^	34.1, CH_2_ ^f^	2.07 m; 0.56 ddd (12.0, 12.0, 3.0) ^e^	34.2, CH_2_ ^f^
2	1.58 m	17.8, CH_2_	1.56 m	17.9, CH_2_
3	1.41 m; 1.17 m	41.8, CH_2_	1.44 m; 1.17 m	41.7, CH_2_
4		33.0, C		33.0, C
5	1.02 m	56.8, CH	0.99 dd (12.6, 2.4)	57.0, CH
6	1.56 m; 1.45 m	18.3, CH_2_	1.58 m; 1.52 m	18.4, CH_2_
7	1.91 m; 1.18 m	41.2, CH_2_	1.92 m; 1.03 m	41.4, CH_2_
8		36.8, C		37.2, C
9	1.41 m	53.5, CH	1.33 m	53.5, CH
10		41.8, C		41.8, C
11	2.35 m; 1.46 m	21.9, CH_2_	1.91 m; 2.25 m	25.1, CH_2_
12	4.97 d (2.8)	76.2, CH	4.96 dd (2.8, 2.8)	76.6, CH
13		41.7, C		41.6, C
14	1.86 dd (13.2, 2.0)	43.6, CH	1.50 m	47.5, CH
15	1.92 m; 2.29 m	25.2, CH_2_	2.14 m; 1.50 m	25.7, CH_2_
16	4.91 dd (4.0, 1.6)	77.2, CH	4.61 dd (8.8, 6.6)	68.1, CH
17		133.6, C		132.4, C
18	6.73 s	155.7, CH	6.59 s	152.4, CH
19	0.85 s	33.8, CH_3_	0.88 s	33.8, CH_3_
20	0.77 s	21.9, CH_3_	0.76 s	21.9, CH_3_
21	1.09 s	16.6, CH_3_	1.10 s	16.4, CH_3_
22	4.05 d (11.6); 3.91 dd (11.6)	62.8, CH_2_	4.27 d (11.4); 3.90 dd (11.4)	62.7, CH_2_
23	1.06 s	19.4, CH_3_	1.20 s	20.9, CH_3_
24		199.3, C		202.1, C
25	2.25 s	25.8, CH_3_	2.25 s	25.7, CH_3_
12-OAc		170.8, C		170.6, C
	2.03 s	21.3, CH_3_	2.04 s	21.3, CH_3_
16-OOH	9.27 s			

^a^ Spectra recorded at 400 MHz in CDCl_3_; ^b^ spectra recorded at 100 MHz in CDCl_3_; ^c^ spectra recorded at 600 MHz in CDCl_3_; ^d^ spectra recorded at 150 MHz in CDCl_3_; ^e^
*J* values (in Hz) in parentheses; ^f^ attached protons were deduced by the HSQC experiment.

**Table 3 pharmaceuticals-16-01258-t003:** Experimental and calculated specific optical rotation values of **1**–**4**.

	Cald. Value ^a^	Exp. Value
Exp. **1** ^b^		21
Cald. **1**-5*S*,8*R*,9*S*,10*R*,13*R*,14*S*,16*S*,17*S*	31	
Cald. **1**-5*R*,8*S*,9*R*,10*S*,13*S*,14*R*,16*R*,17*R*	−31	
Exp. **2** ^c^		61
Cald. **2**-5*S*,8*R*,9*S*,10*R*,12*S*,13*R*,14*S*	78	
Cald. **2**-5*R*,8*S*,9*R*,10*S*,12*R*,13*S*,14*R*	−78	
Exp. **3** ^d^		41
Cald. **3**-5*S*,8*R*,9*S*,10*R*,12*S*,13*R*,14*S*,16*S*	15	
Cald. **3**-5*R*,8*S*,9*R*,10*S*,12*R*,13*S*,14*R*,16*R*	−15	
Exp. **4** ^e^		76
Cald. **4**-5*S*,8*R*,9*S*,10*R*,12*S*,13*R*,14*S*,16*S*	103	
Cald. **4**-5*R*,8*S*,9*R*,10*S*,12*R*,13*S*,14*R*,16*R*	−103	

^a^ Solvent phase in CHCl_3_; ^b^${\;[\alpha ]}_{D}^{25}$ (c 0.025, CHCl_3_); ^c^${\;[\alpha ]}_{D}^{25}$ (c 0.065, CHCl_3_), ^d^$\;{\left[\alpha \right]}_{D}^{25}$ (c 0.045, CHCl_3_); ^e^$\;{\left[\alpha \right]}_{D}^{25}$ (c 0.105, CHCl_3_).

**Table 4 pharmaceuticals-16-01258-t004:** Effect of compounds **1**–**4** on superoxide anion generation and elastase release in fMLF/CB-induced human neutrophils.

Compounds	Superoxide Anion Generation			Elastase Release		
	IC_50_ (μM)	Inh%		IC_50_ (μM)	Inh%	
**1**		9.19 ± 1.06	***		1.59 ± 1.90	
**2**	3.98 ± 0.62	95.02 ± 3.42	***	5.24 ± 0.28	95.47 ± 6.29	***
**3**		12.12 ± 1.22	***		9.70 ± 1.06	***
**4**	4.46 ± 0.72	88.28 ± 5.25	***	4.73 ± 0.40	98.00 ± 4.60	***
**LY294002**	2.66 ± 039	88.30 ± 5.00	***	2.53 ± 0.53	86.71 ± 6.47	***

Percentage of inhibition (Inh.%) at 10 μM. The results are presented as mean ± S.E.M. (n = 3–5). *** *p* < 0.001 compared with the control (DMSO). LY294002 at 10 μM was used as a positive control.

## Data Availability

Data is contained within the article and Appendix A.

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
