# Peer review of "Nor-24-homoscalaranes, Neutrophilic Inflammatory Mediators from the Marine Sponge Lendenfeldia sp."

_pharmaceuticals, 2023, doi:10.3390/ph16091258_

Round 1

Reviewer 1 Report

Congratulations on the very well presented and written paper.

Only very minor comment:
lines 309 and 320: "5" should be in superscript format in  6 × 105 cells/mL

Author Response

Response to Reviewer 1

Congratulations on the very well presented and written paper.

  • We extend our heartfelt gratitude for the thorough reviews and valuable feedback provided by the reviewer. We also express our appreciation for the recognition of our efforts. All the suggestions and comments have been incorporated into our work as part of the revisions.

Only very minor comment: lines 309 and 320: "5" should be in superscript format in 6 × 105 cells/mL

  • We thank the carful comment from the reviewer. We amended the style accordingly. (Page 10; Line 330, 341)

Reviewer 2 Report

This manuscript describes purification and structure determination of four new sesterterpenoids from the marine sponges of the genus Lendenfeldia and their anti-inflammatory activity.

The authors have reported six related compounds from the marine sponges belongs to the same genus in 2020, and now report four new ones.

Structure determination have been appropriately performed by NMR and MS analyses, as well as by comparison with literature data of analogues. Relative and absolute stereochemistry are determined from NOE measurements and calculations of optical rotation. The calculated values correlate well with the experimental values, and are considered reasonable based on previous results of related compounds.

Regarding the biological activity, two compounds showed significant anti-inflammatory activity, whereas two other compounds were almost inactive, and information on structure-activity relationships was also obtained.

Overall, this is an excellent paper on the identification of bioactive natural products. Therefore, we consider the paper acceptable with the following minor modifications.

Are there any known compounds in the crude extract? It is strange that all purified compounds are novel, although many analogous have been reported.

Author Response

Response to Reviewer 2

This manuscript describes purification and structure determination of four new sesterterpenoids from the marine sponges of the genus Lendenfeldia and their anti-inflammatory activity. The authors have reported six related compounds from the marine sponges belongs to the same genus in 2020, and now report four new ones. Structure determination have been appropriately performed by NMR and MS analyses, as well as by comparison with literature data of analogues. Relative and absolute stereochemistry are determined from NOE measurements and calculations of optical rotation. The calculated values correlate well with the experimental values, and are considered reasonable based on previous results of related compounds. Regarding the biological activity, two compounds showed significant anti-inflammatory activity, whereas two other compounds were almost inactive, and information on structure-activity relationships was also obtained. Overall, this is an excellent paper on the identification of bioactive natural products. Therefore, we consider the paper acceptable with the following minor modifications.

  • We extend our heartfelt gratitude for the thorough reviews and valuable feedback provided by the reviewer. We also express our appreciation for the recognition of our efforts. All the suggestions and comments have been incorporated into our work as part of the revisions.

Are there any known compounds in the crude extract? It is strange that all purified compounds are novel, although many analogous have been reported.

  • We appreciate the reviewer's perceptive comments. It is true that we encountered certain replicated scalaranes during the isolation process in our ongoing research. Nonetheless, as the structure elucidation and biological outcomes have already been documented in earlier publications, we have refrained from discussing them in the present manuscript.

Reviewer 3 Report

The authors discovered new anti-inflammatory molecules isolated from the marine sponge Lendenfeldia sp. The authors well-described structure determination of new analogues of lendenfeldaranes via comprehensive NMR data analysis and computational calculation of specific optical rotation values. This article will be interesting for the readers in the marine natural products field. Here are my comments. 

1. Page 2, line 61. Please add full name of IR, SOR, MS, and NMR for non-specialist. 

2. Please correct small typo errors in the main text. For example, in page 3 line 94, H2-6 should be H2-6. 

3. NMR table for compound 3 and 4 should be located in front of Table including SOR calculation. 

3. Figure 3 should be located in front of Table including SOR calculation. Please rearrange this. 

4. Location of NOESY labeling should be improved in Figure 3. 

5. Page 6, line 163. Please explain the meaning of "3H x s". If it is typo error, please correct it. 

6. Page 6, line 169, determination of the position of hydroperoxide group is not fully supported. Please provide clearer explanation. For example, NOESY signal of proton in hydroperoxide group should work. 

7. Please provide background for biological activity assay and the reason for choosing this assay. 

8. Page 8, Table4. Please provide positive control values. 

7. 

Author Response

Response to Reviewer 3

The authors discovered new anti-inflammatory molecules isolated from the marine sponge Lendenfeldia sp. The authors well-described structure determination of new analogues of lendenfeldaranes via comprehensive NMR data analysis and computational calculation of specific optical rotation values. This article will be interesting for the readers in the marine natural products field. Here are my comments.

  • We extend our heartfelt gratitude for the thorough reviews and valuable feedback provided by the reviewer. We also express our appreciation for the recognition of our efforts. All the suggestions and comments have been incorporated into our work as part of the revisions.

  1. Page 2, line 61. Please add full name of IR, SOR, MS, and NMR for non-specialist.
  • We are appreciated for the careful comments from the reviewer. We added the full name accordingly. (Page 2; Line 75-76)

  1. Please correct small typo errors in the main text. For example, in page 3 line 94, H2-6 should be H2-6.
  • We are grateful for the valuable comments. We amended the mentioned typo errors and checked through out the manuscript. (Page 3; Line 109-112; Page 4; Line 127-129)

  1. NMR table for compound 3 and 4 should be located in front of Table including SOR calculation.
  • We thank the kind suggestions from the reviewer. We rearranged the table layouts accordingly.

  1. Figure 3 should be located in front of Table including SOR calculation. Please rearrange this.
  • We thank the kind suggestions from the reviewer. We rearranged the figure and table layouts accordingly.

  1. Location of NOESY labeling should be improved in Figure 3.
  • We would like to thank the comments for the reviews. We improved the resolutions and labeling of figures 3.
  1. Page 6, line 163. Please explain the meaning of "3H x s". If it is typo error, please correct it.
  • We appreciate the careful comment. We corrected the typo accordingly. (Page 7; Line 185)

  1. Page 6, line 169, determination of the position of hydroperoxide group is not fully supported. Please provide clearer explanation. For example, NOESY signal of proton in hydroperoxide group should work.
  • We thank the insight comments from the reviewer. We added the description “Additionally, the NOE correlation originating from H-14 to the hydroxyperoxide proton can also be inferred.” (Page 7; Line 194-195)

  1. Please provide background for biological activity assay and the reason for choosing this assay.
  • We are appreciated for the kind suggestion. We added the background of bioassays in the introduction part “Neutrophils are a type of white blood cell that are essential to the immune response. They play a crucial role in the initial response to infection and tissue injury [46]. While neutrophils are important for combating pathogens and promoting tissue repair, their excessive or prolonged activation can lead to harmful inflammation and tissue damage. Therefore, targeting neutrophil-mediated inflammation is an area of interest in various inflammatory conditions, including autoimmune diseases, acute respiratory distress syndrome (ARDS) [47], and inflammatory bowel disease (IBD) [48]. In our two previous reports, we isolated a series of scalarane-type sesterterpenoids from the marine sponge Lendenfeldia and tested their anti-inflammatory activity against superoxide anion generation and elastase release, which represented the neutrophilic inflammatory responses of respiratory burst and degranulation, respectively. The results indicated the potent activity of this class of compounds to be further developed as anti-neutrophilic agents [5,7].” (Page 2; Line 60-71)

  1. Page 8, Table4. Please provide positive control values.
  • We thank the valuable suggestion. We added the positive control LY294002 in table 4. (Page 8; table 4)

Round 2

Reviewer 3 Report

The authors corrected all issues. 

Author Response

Dear reviewer,

Many thanks for the kind and valuable comments.